# Adversarial Representation Learning for Canonical Correlation Analysis

## Abstract

Canonical correlation analysis (CCA) provides a framework to map multimodality data into a maximally correlated latent space. The deep version of CCA has replaced linear maps with deep transformations to enable more flexible correlated data representations; however, optimization for the CCA target requires calculation on sufficiently large sample batches. Here, we present a deep, adversarial approach to CCA, adCCA, that can be efficiently solved by standard mini-batch training. We reformulate CCA under the assumption that the different modalities are embedded with identical latent distributions, derive a tractable deep CCA target. We implement the new target and distribution constraint with an adversarial framework to efficiently learn the canonical representations. adCCA learns maximally correlated representations across multimodalities meanwhile preserves class information within individual modalities. Further, adCCA removes the need for feature transformation and normalization and can be directly applied to diverse modalities and feature encodings. Numerical studies show that the performance of adCCA is robust to data transformations, binary encodings, and corruptions. Together, adCCA provides a scalable approach to align data across modalities without compromising sample class information within each modality.

## 1 Introduction

Data samples can be measured with different modalities (*e.g.,* image or text), encoded in different formats, and modeled by different distributions. Integrative analysis of multimodality data provides the opportunity in many machine learning tasks to combine partial information from each modality and achieve better performance than any single modality alone (Ngiam et al. (2011); Srivastava & Salakhutdinov (2012)). Canonical correlation analysis (CCA) (Thompson (1984)) is one of the most classical and general approaches for multimodality data integration. It learns linear mappings between data modalities that achieve maximal cross-modality correlations.

Replacing the linear mappings in CCA with deep functions can achieve non-linear and flexible transformations and provide better correlated representations. However, learning the deep CCA target function requires optimization over all data samples (Andrew et al. (2013)) or a sufficiently large data batch (Wang et al. (2015)), which is incompatible with standard batch-based learning strategies widely used in deep learning and limits its power on large-scale datasets. Therefore, many recent deep CCA approaches (Wang et al. (2016); Dutton (2020); Karami & Schuurmans (2021)) sidestep the original CCA formulation and rather focus on learning a joint representation for the paired modalities using approaches that are compatible with mini-batch training (Appendix A).

Here, we propose a multimodal adversarial learning framework for deep CCA learning: adCCA. Mathematical analysis provides an optimization target for CCA amenable to mini-batch training under the requirement that the different modality distributions are identical in latent space. adCCA formulates this requirement as a penalty function that, during optimization, brings the two latent distributions into alignment. As the latent space representations converge (distribution-wise), maximizing the optimization target leads to highly correlated latent representations (sample-wise) for the two modalities, as illustrated with numerical experiments. Thus, adCCA is derived from a deep CCA framework that follows the original correlation target of CCA, yet can be directly optimized by mini-batch training.

In more detail, adCCA learns representations in multiple steps. First, initial latent representations are provided using modality-specific autoencoders (AEs). Second, the Wasserstein distance is used to measure overall differences between the two latent distributions, and an inner product is used to measure sample-wise similarities. Then, adCCA uses an adversarial framework to minimize latent space distribution differences and maximize sample-wise similarity (Goodfellow et al. (2014)). Together, the final output representation from each modality is expected to have similar latent distributions (from the Wasserstein distance terms), high cross-modality sample similarity (from the inner product term) while still maintaining a faithful representation of the original features (from the AE).

The design of adCCA framework has several advantages. One major advantage is flexibility in input features. Often in multimodal analysis, the total loss is composed of loss terms from different modalities (Wan et al. (2021); Andrew et al. (2013); Ngiam et al. (2011); Jain et al. (2005)). If features from diverse modalities have different distributions or data formats, the joint optimization can be biased towards a certain modality. In the adCCA framework, each modality representation is generated from a modality-specific encoder, and the two latent representations are only related indirectly through the adversarial network. Another major advantage is the ability to learn correlation without losing information in the original features. In particular, deep CCA approaches use deep transformations to learn canonical representations with high correlation; however, in doing so these representations may lose class information contained in the original feature space. This point is typically not considered in deep CCA frameworks. In adCCA, the use of AE in learning forces the latent representation to retain a faithful reconstruction of the original data.

We benchmark adCCA against 10 different methods, using both synthetic and real datasets. In the evaluation, we focused on two aspects. *Cross-modality consistency*: Correlation of sample points in the latent space for both modalities (Appendix Fig D.1B). And, *Class preservation*: Ability of the latent representation to retain the original class information (Appendix Fig D.1C). The ideal method should provide embedding that are both correlated per sample and retain class structure across both modalities (Appendix Fig D.1D). We show representations from adCCA both achieve high cross-modality consistency in the joint latent space and preserve class information from the original feature distributions. We demonstrate that adCCA can maintain stable performances across features of different modality types, feature distributions, and degradation levels.

The major contributions of this work are: (1) A reformulation of CCA with deep transformations into a target that can be optimized through standard mini-batch training. (2) An adversarial learning framework for efficient canonical representation learning and an optimization strategy for the stable adversarial training between two latent representations. (3) An expanded approach for evaluating the performance of CCA approaches for both cross-modality consistency and single-modality information preservation. (4) An extensive test and demonstration of adCCA's performance.

## 2 ADVERSARIAL REPRESENTATION LEARNING FOR CCA

### 2.1 PRELIMINARY

Here, we consider a general framework for CCA. (An overview of existing CCA variants is provided in Appendix A.) Given a pair of feature vectors from two modalities, $\mathbf{x} \in \mathbb{R}^p$ and $\mathbf{y} \in \mathbb{R}^q$, CCA tries to learn two mapping functions $f_1$ and $f_2$ that maximize the correlation after mapping:

$$\begin{aligned} &\max corr(z_x, z_y) \\ &\text{s.t. } z_x = f_1(\mathbf{x}), z_y = f_2(\mathbf{y}), \end{aligned} \tag{1}$$

where $z_x \in \mathbb{R}$ and $z_x \in \mathbb{R}$ are the corresponding canonical variables. Transformation functions $f_1$ and $f_2$ can be linear mappings (in the original CCA), implicit kernel functions (in kernel CCA) or deep transformations (in deep CCA). These functions are learned through optimization over the entire dataset (Andrew et al., Appendix B.1), and mini-batch training strategies cannot be leveraged.

Here, we aim to reformulate the CCA learning target to a form that enables optimization through general deep-learning optimizers. To this end, we relax the CCA learning target with an additional identical latent distribution assumption:

**Theorem 1.** *Assuming latent representations from modality $x$ and $y$ follow the same distribution $\pi$ with the mean $\mu$ and variance $\sigma^2$, the optimization target of CCA (Eq. 1) can be rewritten as:*

$$\max \mathbb{E}_{(\mathbf{x},\mathbf{y})\sim p_{\text{data}}}[z_x z_y]$$
$$\text{s.t. } z_x = f_1(\mathbf{x}), z_y = f_2(\mathbf{y}), \tag{2}$$
$$z_x \sim \pi(\mu,\sigma^2), z_y \sim \pi(\mu,\sigma^2),$$

*where $(\mathbf{x},\mathbf{y}) \sim p_{\text{data}}$ indicates paired modal features $\mathbf{x}$ and $\mathbf{y}$ drawn from the data.*

*Proof.* Refer to Appendix B.2. $\qquad\square$

The latent distributions are assumed to be the same with the same fixed $\mu$ and $\sigma^2$, and the new target function can be expressed simply as the average of the product of canonical variables. This formulation avoids noted problems with mini-batch optimization in CCA where $\mu$ and $\sigma^2$ are not fixed.

We next extend the optimization target to learn $k$ canonical variables simultaneously. Deep CCA (Appendix B.1) employs Singular Value Decomposition (SVD) to output canonical variables that are orthogonal to each other. This step requires the use of large data batch sizes. To achieve more efficient optimization, we forgo the requirement for orthogonality. Instead, we use two independent autoencoders to generate canonical representations. We add the autoencoders as constraint terms and extend Theorem 1 to a multi-variable form:

**Corollary 1.** *Multidimensional canonical representations $\mathbf{z}_x \in \mathbb{R}^k$ and $\mathbf{z}_y \in \mathbb{R}^k$ are learned from two individual autoencoders with encoders $f_1(\mathbf{x})$ and $f_2(\mathbf{y})$ and decoders $g_1(\mathbf{z}_x)$ and $g_2(\mathbf{z}_y)$. Under the assumption that each entry of the two latent representations $\mathbf{z}_x^{(j)}$, $\mathbf{z}_y^{(j)}$ follow the same latent distribution with mean $\mu$ and variance $\sigma^2$, the optimization target of CCA can be re-formulated as:*

$$\max \mathbb{E}_{(\mathbf{x},\mathbf{y})\sim p_{\text{data}}}[\mathbf{z}_x^T \mathbf{z}_y]$$
$$\text{s.t. } \mathbf{z}_x = f_1(\mathbf{x}), \mathbf{z}_y = f_2(\mathbf{y}),$$
$$\|\mathbf{x} - g_1(\mathbf{z}_x)\|_2^2 = 0, \|\mathbf{y} - g_2(\mathbf{z}_y)\|_2^2 = 0 \tag{3}$$
$$\mathbf{z}_x^{(j)}, \mathbf{z}_y^{(j)} \sim \pi(\mu,\sigma^2) \text{ for } j = 1 : k$$

*Proof.* Refer to Appendix B.3. $\qquad\square$

## 2.2 MODEL FORMULATION

Based on Corollary 1, the new target function for CCA has three requirements: 1) maximizing the inner product between latent representations of the two modalities; 2) requiring the two latent distributions from two modalities to be the same; and 3) fixing the mean and variance of each latent dimension to be constant.

These requirements can be easily satisfied using existing deep-learning paradigms. For (1), the inner product is a general term to measure sample similarities and is widely used in deep-learning approaches, such as simCLR (Chen et al. (2020)); for (2), distribution differences can be measured by the Wasserstein distance (Arjovsky et al. (2017)) or by other types of divergences used in machine learning tasks, such as image generation (Makhzani et al. (2015)); for (3), constant mean and variance can be achieved through batch normalization.

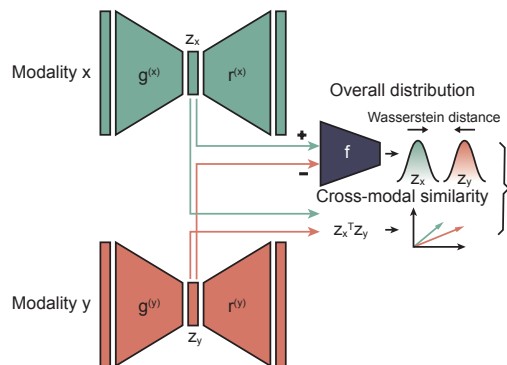

Figure 1: adCCA framework and loss function design.

Next, we describe the adCCA framework (Fig. 1; Appendix C). Here, we consider paired modality features $\mathbf{x}$ and $\mathbf{y}$ collected from the same sample. First, we employ an autoencoder (AE) for each modality to learn the initial latent representations $\mathbf{z}_x$ and $\mathbf{z}_y$:

$$
\begin{aligned}
\min_{\varphi_x, \theta_x} & \ \mathbb{E}_{\mathbf{x} \sim p_x} \| r_{\varphi_x}^{(x)} [g_{\theta_x}^{(x)}(\mathbf{x})] - \mathbf{x} \|_2^2, \\
\min_{\varphi_y, \theta_y} & \ \mathbb{E}_{\mathbf{y} \sim p_y} \| r_{\varphi_y}^{(y)} [g_{\theta_y}^{(y)}(\mathbf{y})] - \mathbf{y} \|_2^2,
\end{aligned}
\tag{4}
$$

where $g_{\theta_x}^{(x)}$ and $r_{\varphi_x}^{(x)}$ are the encoder and decoder for modality $x$ with network parameter sets $\theta_x$ and $\varphi_x$ (*Resp.* $g_{\theta_y}^{(y)}$ and $r_{\varphi_y}^{(y)}$ for modality $y$); $p_x$ and $p_y$ are data distributions for $\mathbf{x}$ and $\mathbf{y}$. The latent representations are given from the modality-specific encoders by $\mathbf{z}_x = g_{\theta_x}^{(x)}(\mathbf{x})$ and $\mathbf{z}_y = g_{\theta_y}^{(y)}(\mathbf{y})$. The reconstruction loss encourages the AE bottleneck layer to faithfully encode information contained in the original feature space.

Second, a Wasserstein distance is used to measure the distance between the latent distributions of the two modalities:

$$
\begin{aligned}
W(p_{\mathbf{z}_x}, p_{\mathbf{z}_y}) & = \inf_{\gamma_z \in \Pi(p_{\mathbf{z}_x}, p_{\mathbf{z}_y})} \mathbb{E}_{(\mathbf{z}_x, \mathbf{z}_y) \sim \gamma_z} [\| \mathbf{z}_x - \mathbf{z}_y \|] \\
& = \inf_{\gamma \in \Pi(p_x, p_y)} \mathbb{E}_{(\mathbf{x}, \mathbf{y}) \sim \gamma} [\| \mathbf{z}_x - \mathbf{z}_y \|] \\
& \text{s.t. } \mathbf{z}_x = g_{\theta_x}^{(x)}(\mathbf{x}), \mathbf{z}_y = g_{\theta_y}^{(y)}(\mathbf{y}),
\end{aligned}
\tag{5}
$$

where $p_{\mathbf{z}_x}, p_{\mathbf{z}_y}$ are distributions for $\mathbf{z}_x$ and $\mathbf{z}_y$, and $\Pi(p_{\mathbf{z}_x}, p_{\mathbf{z}_y})$ represents the space of joint distributions for $\mathbf{z}_x$ and $\mathbf{z}_y$ with marginal distributions $p_{\mathbf{z}_x}$ and $p_{\mathbf{z}_y}$. The same notation also applies to $\Pi(p_x, p_y)$. As previously shown (Arjovsky et al. (2017)), the least upper bound of Wasserstein distance can also be formulated in terms of a 1-Lipschitz function $f_w$:

$$
W(p_{\mathbf{z}_x}, p_{\mathbf{z}_y}) = \sup_{\| f_w \|_L \leq 1} \left( \mathbb{E}_{\mathbf{x} \sim p_x} [f_w(\mathbf{z}_x)] - \mathbb{E}_{\mathbf{y} \sim p_y} [f_w(\mathbf{z}_y)] \right).
\tag{6}
$$

Following the Wasserstein GAN structure (Arjovsky et al. (2017)), we minimize this distance in an adversarial paradigm:

$$
\begin{aligned}
\max_w \min_{\theta_x, \theta_y} & \ \left( \mathbb{E}_{\mathbf{x} \sim p_x} [f_w(\mathbf{z}_x)] - \mathbb{E}_{\mathbf{y} \sim p_y} [f_w(\mathbf{z}_y)] \right) \\
& \text{s.t. } \mathbf{z}_x = g_{\theta_x}^{(x)}(\mathbf{x}), \mathbf{z}_y = g_{\theta_y}^{(y)}(\mathbf{y}).
\end{aligned}
\tag{7}
$$

Here, the AE encoders $g_{\theta_x}^{(x)}$ and $g_{\theta_y}^{(y)}$ from above are reused as generative networks. $f_w$ is modeled by a discriminator network with parameter set $w$. Maximizing the function by optimizing the discriminator parameter $w$ increases the difference between two representations $\mathbf{z}_x$ and $\mathbf{z}_y$, while minimizing the function by tuning generator parameters decreases the distance by finding better encoder representations.

Third, we add the inner product to Eq. 7 and formulate the learning target used by adCCA:

$$
\begin{aligned}
\max_w \min_{\theta_x, \theta_y} & \ -\mathbb{E}_{(\mathbf{x}, \mathbf{y}) \sim p_{\text{data}}} [\mathbf{z}_x^T \mathbf{z}_y] + \lambda \left[ \mathbb{E}_{\mathbf{x} \sim p_x} [f_w(\mathbf{z}_x)] - \mathbb{E}_{\mathbf{y} \sim p_y} [f_w(\mathbf{z}_y)] \right], \\
& \text{s.t. } \mathbf{z}_x = g_{\theta_x}^{(x)}(\mathbf{x}), \mathbf{z}_y = g_{\theta_y}^{(y)}(\mathbf{y}),
\end{aligned}
\tag{8}
$$

where $\lambda > 0$ is a multiplier used to balance the distribution penalty (given by the Wasserstein distance) with the sample pairwise-similarity term (given by the inner product of latent representations). Taken together, the overall adCCA model requires the optimization of Eqs. 4 and 8.

We note that the two latent representations are provided from two modality-specific AEs and are not linked directly in the model. Thus, there is no requirement to process raw features $\mathbf{x}$ and $\mathbf{y}$ for balanced learning and the two AEs can be replaced with modality specific structure (*e.g.,.* one AE can be a convolutional network for images and the other an attention model for language).

### 2.3 INTUITIVE INTERPRETATION OF THE ADCCA LOSS FUNCTION

In this section, we provide intuition for how the formulated loss function can provide correlated embeddings from multimodal data. In Eq. 8, the first two Wasserstein distance terms force the two latent representations, $\mathbf{z}_x$ and $\mathbf{z}_y$, to share similar distribution shapes. However, this overall distribution penalty does not align different modality features from the same sample. The last inner product term explicitly encourages representations generated from paired modality features corresponding to the same samples to maximize their similarities. Together, adCCA encourages similarity across modalities in the latent space representations at both distribution (Wasserstein distance) and sample (feature correlation) scales.

The AEs in Eq. 4 encourage $\mathbf{z}_x$ and $\mathbf{z}_y$ to preserve information from each modality. This allows adCCA to avoid the adversarial network converging to trivial solutions. For example, $\mathbf{z}_x$ and $\mathbf{z}_y$ could converge to zero vectors, which satisfy both the distribution and similarity requirements, but do not carry any information from the original data. By using both Eqs. 4 and 8, multimodal representations can be learned that both preserve the original information and achieve high cross-modality correlations.

### 2.4 MODEL OPTIMIZATION

In the standard generative adversarial network framework (Goodfellow et al. (2014); Arjovsky et al. (2017)), we only need to optimize one generator and discriminate its output against real samples. However, in adCCA, we need to optimize two generative networks and match the distributions between both latent representations. During training, these two latent space representations change epoch by epoch, which creates a challenge for a stable optimization.

Here, we propose a multi-step training process for stable adCCA optimization in Algorithm 1 (illustrated in Appendix Fig. D.2). First, autoencoders are trained independently for each modality to provide an initial embedding. This enables adCCA to work with raw input feature data from different modalities and different distributions (see Section 3.1.2). Next, we fix the representation for modality

---

**Algorithm 1:** adCCA optimization

**Data:** $\{(\mathbf{x}, \mathbf{y})\}^n \sim p_{\text{data}}, \mathbf{x} \in \mathbb{R}^p, \mathbf{y} \in \mathbb{R}^q$
**Result:** $\{(\mathbf{z}_x, \mathbf{z}_y)\}^n, \mathbf{z}_x \in \mathbb{R}^k, \mathbf{z}_y \in \mathbb{R}^k$
`# Initialize` $\mathbf{z}_x$ `and` $\mathbf{z}_y$ `with autoencoders`
**for** $i \leftarrow 1 : \text{epoch}^{(\text{AE})}$ **do**

$\quad \varphi_x, \theta_x \leftarrow \min \mathbb{E}_{\mathbf{x} \sim p_x} \| r_{\varphi_x}^{(x)} [g_{\theta_x}^{(x)}(\mathbf{x})] - \mathbf{x} \|_2^2;$

$\quad \varphi_y, \theta_y \leftarrow \min \mathbb{E}_{\mathbf{y} \sim p_y} \| r_{\varphi_y}^{(y)} [g_{\theta_y}^{(y)}(\mathbf{y})] - \mathbf{y} \|_2^2;$

**end**
$\mathbf{z}_x = g_{\theta_x}^{(x)}(\mathbf{x}), \mathbf{z}_y = g_{\theta_y}^{(y)}(\mathbf{y});$
`# Optimize` $\mathbf{z}_x$ `and` $\mathbf{z}_y$ `iteratively`
**for** $i \leftarrow 1 : \text{epoch}^{(\text{AD})}$ **do**

$\quad$ `#` $\underline{y\text{-step:}}$ `fix` $\mathbf{z}_x$ `and optimize` $\mathbf{z}_y$

$\quad \varphi_y, \theta_y \leftarrow \min \mathbb{E}_{\mathbf{y} \sim p_y} \| r_{\varphi_y}^{(y)} [g_{\theta_y}^{(y)}(\mathbf{y})] - \mathbf{y} \|_2^2;$

$\quad w \leftarrow \max \mathbb{E}_{\mathbf{x} \sim p_x} [f_w(\mathbf{z}_x)] - \mathbb{E}_{\mathbf{y} \sim p_y} [f_w(\mathbf{z}_y)];$

$\quad \theta_y \leftarrow \min - \mathbb{E}_{\mathbf{y} \sim p_y} [f_w(\mathbf{z}_y)] - \lambda \mathbb{E}_{(\mathbf{x},\mathbf{y}) \sim p_{\text{data}}} [\mathbf{z}_x^T \mathbf{z}_y];$

$\quad$ `#` $\underline{x\text{-step:}}$ `fix` $\mathbf{z}_y$ `and optimize` $\mathbf{z}_x$

$\quad \varphi_x, \theta_x \leftarrow \min \mathbb{E}_{\mathbf{x} \sim p_x} \| r_{\varphi_x}^{(x)} [g_{\theta_x}^{(x)}(\mathbf{x})] - \mathbf{x} \|_2^2$

$\quad w \leftarrow \max \mathbb{E}_{\mathbf{x} \sim p_x} [f_w(\mathbf{z}_x)] - \mathbb{E}_{\mathbf{y} \sim p_y} [f_w(\mathbf{z}_y)];$

$\quad \theta_x \leftarrow \min \mathbb{E}_{x \sim p_x} [f_w(\mathbf{z}_x)] - \lambda \mathbb{E}_{(\mathbf{x},\mathbf{y}) \sim p_{\text{data}}} [\mathbf{z}_x^T \mathbf{z}_y];$

**end**
**return** $\mathbf{z}_x = g_{\theta_x}^{(x)}(\mathbf{x}), \mathbf{z}_y = g_{\theta_y}^{(y)}(\mathbf{y})$

---

$x$ and optimize the generative network for modality $y$ through the adversarial learning step ($y$-step). During this $y$-step, we optimize the autoencoder, discriminator and generator. The discriminator identifies distributional differences, which are then minimized by "moving" the latent distribution of $y$ towards the latent distribution of $x$. Meanwhile, the autoencoder ensures that the latent representation faithfully reflects information in the original data. Then, we fix the representation of $y$ and optimize the generator for $x$ ($x$-step). The $x$-step and $y$-step are performed iteratively.

## 3 EXPERIMENTS

In this section, we show the results of experiments using simulated and real data. We compared the proposed adCCA with other CCA approaches as well as multimodal learning frameworks that can complete the same task.

Specifically, we compared adCCA to two different classes of existing approaches. The first class (individual representation learning methods) provides an individual representation for each modality. This class contains 6 approaches: classical CCA, kernel CCA (radial basis function and polynomial kernel) (Perry et al. (2021)), CCA with deep transformations (DCCA) (Andrew et al. (2013)), simCLR (Chen et al. (2020)) that employs the same encoder to learn similar representations from multimodal inputs, and contrastive autoencoders (contrast AE) that uses a contrastive loss to constrain the similarities in the latent space (which can be viewed as adCCA without the distribution penalty). The second class (joint representation learning methods) provided a combined representation for both modalities. This class contained 4 approaches: CCA with the latent distribution constraint under variational framework (VCCA and VCCA-p with private latents) (Wang et al. (2016)) or adversarial framework (ACCA) (Dutton (2020)), and deep probabilistic CCA (DPCCA) (Karami & Schuurmans (2021)). Detailed configurations were provided in Appendix E.1.

### 3.1 EXPERIMENTS WITH SYNTHETIC DATA

We first performed simulations studies by generating synthetic multimodal features with known sample labels. Here, we considered 10 ground truth classes existing in the data and generated synthetic features using multi-layer perception networks with random mapping matrices for the two modalities (Appendix E.2). Then, we used multimodal approaches to learn representations in latent spaces of 10 dimensions.

### 3.1.1 BASELINE COMPARISONS

We set high feature drop-out rates for features in modality $x$ (90% feature missing ratio and 0.1 noise variance) and high noise levels for modality $y$ (10% feature missing ratio and 0.5 noise variance). We observed, based on visualization of the raw feature spaces using Umap (Becht et al. (2019)) (Fig. 2, left), that the high drop-out rates in $x$ had more significant effects on degrading the data class information than the high noise levels in $y$. We then tested the ability to map both modalities to the same latent space.

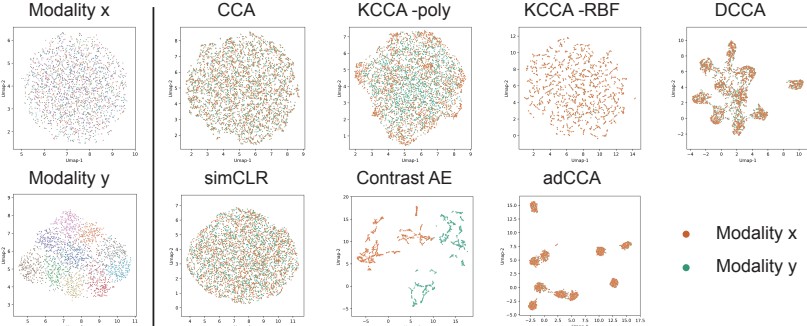

Figure 2: Evaluation of cross-modality consistency. Left: visualization of raw features from two modalities using Umap (Becht et al. (2019)). Color indicates the sample classes. Right: joint visualization of two modalities in the same latent space. Color indicates the modality that representation was learned from. KCCA-poly: CCA with polynomial kernel; KCCA-RBF: CCA with radial basis function kernel.

First, we evaluated the degree of correlation between the latent representations of the two modalities across different methods. We co-embedded the two modal representations using Umap. (We note that ACCA, VCCA(-p) and DPCCA can only output a joint representation and were not evaluated for this comparison.) For most approaches, representations from different modalities were mixed in the latent space (Fig. 2, right). Quantitative assessment in 10-dimension latent spaces using correlation coefficients and entropy of mixing (Appendix E.3) showed that CCA, DCCA and the proposed adCCA had reasonable performance for combining representations from different modalities (Table 1). Kernel CCA also achieved good performance with appropriate kernel choices (RBF kernel in this case).

Second, we evaluated whether the representations of the two modalities in the latent space reflected sample clusters present in the original data. It was evident from 2D representation embeddings with the original class labels (Fig. 3) that DCCA and adCCA were best suited to preserve cluster

| Method | Correlation coefficient | Entropy of mixing |
|---|---|---|
| CCA | $0.976 \pm 0.002$ | $0.682 \pm 0.002$ |
| KCCA-poly | $0.732 \pm 0.009$ | $0.512 \pm 0.158$ |
| KCCA-RBF | $0.978 \pm 0.001$ | $\mathbf{0.693 \pm 0.001}$ |
| DCCA | $0.985 \pm 0.009$ | $0.689 \pm 0.005$ |
| simCLR | $0.335 \pm 0.102$ | $0.688 \pm 0.007$ |
| Contrast AE | $-0.074 \pm 0.396$ | $0.185 \pm 0.234$ |
| adCCA | $\mathbf{0.987 \pm 0.011}$ | $0.690 \pm 0.004$ |

Table 1: Evaluation of cross-modality consistency for latent representations learned from two modalities. Correlation coefficients were calculated on each latent dimension and their mean and standard deviation were reported.

| Method | ARI | Silloutte | SVM | $k$NN |
|---|---|---|---|---|
| CCA | 0.019 | -0.03 | $0.114 \pm 0.017$ | $0.370 \pm 0.008$ |
| KCCA-poly | 0.273 | 0.018 | $0.121 \pm 0.033$ | $0.457 \pm 0.174$ |
| KCCA-RBF | 0.002 | -0.020 | $0.107 \pm 0.002$ | $0.233 \pm 0.018$ |
| DCCA | 0.641 | 0.225 | $0.905 \pm 0.007$ | $0.902 \pm 0.008$ |
| simCLR | 0.012 | -0.037 | $0.201 \pm 0.059$ | $0.128 \pm 0.014$ |
| Contrast AE | 0.034 | -0.041 | $0.294 \pm 0.032$ | $0.318 \pm 0.020$ |
| ACCA | 0.462 | 0.113 | $0.408 \pm 0.004$ | $0.681 \pm 0.028$ |
| VCCA | 0.451 | 0.062 | $0.929 \pm 0.011$ | $0.793 \pm 0.022$ |
| VCCA-p | 0.014 | -0.049 | $0.128 \pm 0.023$ | $0.149 \pm 0.015$ |
| DPCCA | 0.008 | -0.080 | $0.127 \pm 0.018$ | $0.140 \pm 0.007$ |
| adCCA | $\mathbf{0.924}$ | $\mathbf{0.237}$ | $\mathbf{0.991 \pm 0.003}$ | $\mathbf{0.975 \pm 0.005}$ |

Table 2: Evaluation of preserving class information in the latent space. Classification accuracy was based on 5-fold cross validations.

structure in the original data. We quantified the quality of the latent representations to reflect true sample classes using three measures: 1) clustering the latent representations and comparing cluster identities with the true class labels using the adjusted Rand index (ARI); 2) calculating between- and within-class distances using the true labels and quantifying sample compactness using the Silhouette coefficient; and 3) constructing simple classifiers in the latent space using the true labels and reporting classification accuracy. Details of these measures are provided in Appendix E.3.

The results of these evaluations (Table 2) confirmed that DCCA and adCCA performed well at retaining class information from the original data. While some of the other approaches could align multimodal representations in the latent space (i.e., high correlation; Table 1), they did so at the cost of losing spatial coherency present in the original features (Fig. 3,Table 2). In general, higher correlation need not lead to better class predictions for classical approaches as their only learning target is the maximization of the correlation between two representations. To get a high correlation, the feature information from original data can be greatly compromised (An illustration in Appendix Fig. D.1).

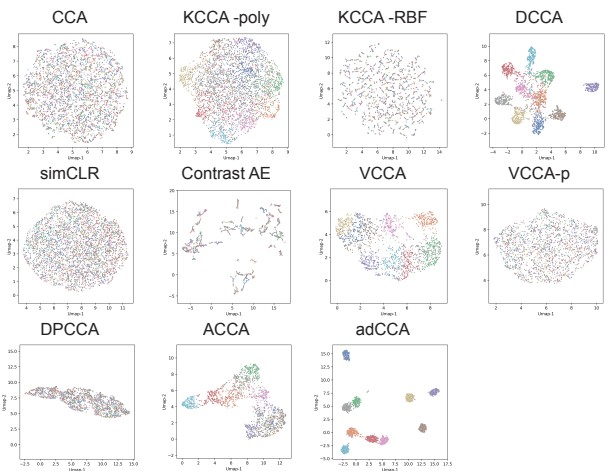

Figure 3: Evaluation of preserving original data classes in latent space. Umap embeddings of latent representations are as in Fig. 2, with color annotations indicating the true sample labels.

Overall, our experiments suggest that the adCCA framework can achieve a balance between cross-modality consistency (encouraged by the adversarial loss) and single-modality preservation (encouraged by the autoencoder loss). We provide a 2D illustration of how adCCA gradually aligns the two modalities in the latent space during training (Appendix D.3) as well as a systematic structure test to highlight the impact of the different key components of the adCCA model architecture in achieving high correlation and classification prediction (Appendix Table F.1).

### 3.1.2 PERFORMANCE WITH INPUT TRANSFORMATIONS

A known challenge in the integrative analysis of multimodal data is balancing contributions from different modalities, especially when data are encoded in different formats or the data distributions are unknown. In the adCCA framework, to enable flexibility of input encoding, independent AEs are used to construct latent encodings for each modality, then the representations are aligned *indirectly* using an adversarial network.

To test the stability of performance, we considered two types of transformations to the raw features $y$ used in the previous experiments: 1) a feature distribution transformation, by applying two sequential $\log$ transformations on entry values; and 2) a feature encoding transformation, by mapping features into binary codes with local sensitive hashing (Appendix E.2). We evaluated how these transformations changed the consistency between the two modal representations (*i.e.*, correlation) as well as the preservation of original data classes (*i.e.*, classification accuracy).

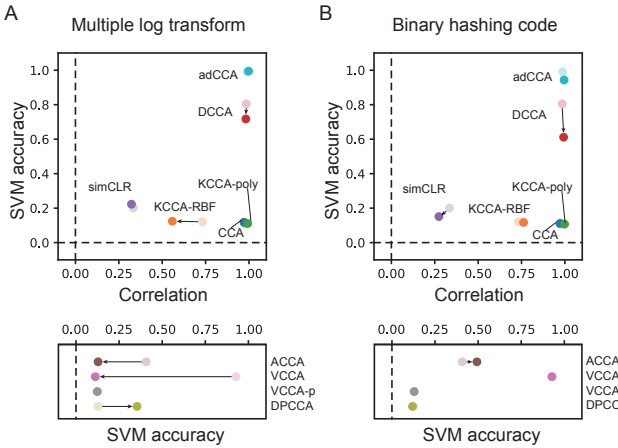

Figure 4: Evaluation of performance after (A) $\log$ or (B) binary hashing transform of the input $y$. Top: Analysis of individual representation learning methods. Axes: consistency (*i.e.*, correlation) and class prediction (*i.e.*, accuracy). Bottom: Analysis of joint representation learning methods. Axis: class prediction. Colors: lighter or darker colors represent results before or after transforms (*resp.*). Arrows: large changes.

From Fig. 4, we observed that some methods had stable class prediction scores but unstable correlation (*e.g.*, DCCA), while others had unstable class prediction scores but stable correlation (*e.g.*, KCCA-RBF). A few were stable under all tested measurements (*e.g.*, adCCA, CCA, KCCA-poly, VCCA-p). The joint representation learning approaches (ACCA, VCCA and DPCCA) showed strong accuracy change under log transform, implying different distributions of multimodal features can affect the effective information combination. Notably, adCCA was stable under both transforms and performed well overall (Fig. 4, in upper right quadrants of top scatter plots). Full results were provided in the Appendix Tables F.2, F.3, F.4 and F.5 .

## 3.2 EXPERIMENTS WITH REAL DATA

Next, we performed evaluations on real datasets with multimodal features. We considered a multiview MNIST data where two views of the digits were related through transformations and a CITE-seq multimodal dataset on spleen and lymph node where gene and protein features were measured directly from single cells.

### 3.2.1 MNIST: DIGITS WITH TWO TRANSFORMED VIEWS

For each digit, we applied random image transformations (rotation, translation, scaling, shearing and random pixel corruption) to generate two views for the same digit image (Appendix E.4). We then learned single or joint representations for the modalities. We benchmarked how representations from the two views mixed in the latent space and how these representations reflected different digit classes.

From the results (Fig. 5A), we found both the classical and kernel CCA approaches showed compromised performance. DCCA and adCCA still maintained high consistency between the two views as well as class information. Further, the runtime (Appendix Fig. D.4) showed adCCA was shower than VCCA, DPCCA and simCLR due to its multiple step training, but faster than classical approaches KCCA, DCCA. We selected these two top approaches and stress-tested them with expanded transformation ranges of 2-fold or 3-fold (Appendix E.4). Under these increasing degradation levels, adCCA exhibited the most robust performance with less loss in correlation and accuracy (Fig. 5b).

Figure 5: Integrative analysis of two transformed views from MNIST dataset.

(a) Overall performance

| Method | Correlation | Class prediction |
|---|---|---|
| CCA | $0.544 \pm 0.204$ | $0.412 \pm 0.000$ |
| KCCA-poly | $0.412 \pm 0.012$ | $0.325 \pm 0.004$ |
| KCCA-RBF | $0.528 \pm 0.073$ | $0.409 \pm 0.001$ |
| DCCA | $\mathbf{0.991 \pm 0.008}$ | $0.601 \pm 0.002$ |
| simCLR | $0.845 \pm 0.064$ | $0.462 \pm 0.001$ |
| Contrast AE | $0.432 \pm 0.054$ | $0.475 \pm 0.032$ |
| ACCA | - | $0.505 \pm 0.007$ |
| VCCA | - | $0.508 \pm 0.006$ |
| VCCA-p | - | $0.409 \pm 0.003$ |
| DPCCA | - | $0.523 \pm 0.042$ |
| adCCA | $0.970 \pm 0.016$ | $\mathbf{0.713 \pm 0.002}$ |

(b) Increased data degradations

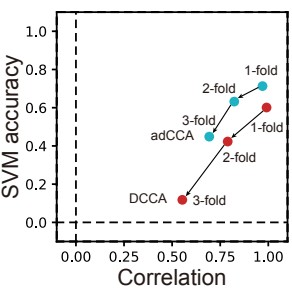

### 3.2.2 CITE-SEQ: CELLS WITH TWO MODALITY PROFILES

Finally, we tested performance on human spleen and lymph node cells measured by CITE-seq (Gayoso et al. (2021)). This technology profiles genes and proteins simultaneously from the same cells. After data preprocessing 9,264 cells (Appendix E.5), modality $x$ includes 2,000 genes and modality $y$ include 112 proteins. It has been reported that the gene data follow a negative binomial distributions, while the protein data follow a negative binomial mixture distribution (Gayoso et al. (2021)). Transformations on raw data may reduce biological signals. Therefore, we evaluated how well the different computational methods could directly integrate these two types of data.

| Method | Correlation | SVM prediction | |
|---|---|---|---|
| | | 5 cell types | 35 subtypes |
| CCA | $0.877 \pm 0.070$ | $0.481 \pm 0.001$ | $0.295 \pm 0.002$ |
| KCCA-poly | $0.985 \pm 0.007$ | $0.413 \pm 0.002$ | $0.225 \pm 0.003$ |
| KCCA-RBF | $0.983 \pm 0.000$ | $0.523 \pm 0.000$ | $0.315 \pm 0.002$ |
| DCCA | $\mathbf{0.994 \pm 0.004}$ | $0.819 \pm 0.001$ | $0.738 \pm 0.003$ |
| simCLR | - | - | - |
| Contrast AE | $0.452 \pm 0.003$ | $0.643 \pm 0.030$ | $0.389 \pm 0.012$ |
| ACCA | - | $0.917 \pm 0.003$ | $0.681 \pm 0.005$ |
| VCCA | - | $0.919 \pm 0.003$ | $0.766 \pm 0.006$ |
| VCCA-p | - | $0.601 \pm 0.009$ | $0.496 \pm 0.004$ |
| DPCCA | - | $0.903 \pm 0.005$ | $0.653 \pm 0.010$ |
| adCCA | $0.988 \pm 0.015$ | $\mathbf{0.930 \pm 0.001}$ | $\mathbf{0.781 \pm 0.005}$ |

Table 3: Integrative analysis of gene and protein modalities from CITE-seq. Prediction results were based on 5-fold cross validation using linear SVM. We note that simCLR could not be evaluated because of the unequal modal dimensions.

We benchmarked using correlations of multimodal representations as well as their prediction accuracies on major classes (5) and subclasses (35) of cell types (Table 3). For correlation, most CCA approaches showed reasonable performance in consistency of latent embeddings. However, approaches outputting joint representations (ACCA and VCCA) had better prediction performance. adCCA maintained both high correlation and prediction accuracy.

## 4 DISCUSSION AND FUTURE WORKS

In this work, we formulated the CCA task under the identical latent distribution assumption and derived an adCCA framework that can be efficiently solved by standard mini-batch optimizers. The training of adCCA includes both cross-modality consistency constraint and single-modality information preservation. Analysis of both simulated and real datasets suggests that adCCA provides an effective approach to learn correlated latent representations and maintain original class information. Compared with the original CCA framework, adCCA has several limitations and directions for future exploration. First, adCCA's latent representations are output from independent encoders and therefore cannot maintain orthogonality of representations. Second, adCCA's latent representations are limited by the AE framework, though the AE can be replaced with more general (such as attention-based model) or modality-specific representation learning frameworks to improve performance. Third, adCCA training involves simultaneously optimization of the GAN and AE, which can be time-consuming to reach to a balanced state. Finally, the use of inner products to constrain similarities between modalities in adCCA may be replaced by error models (such as $L_2$ or Huber loss) to enforce stronger similarity constraints. Nevertheless, adCCA provides a starting for for scalable alignment of data across modalities that preserves structure within each modality.

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

APPENDIX

## A  RELATED WORKS

In the development of CCA, early improvements (Lai & Fyfe (2000); Hardoon et al. (2004)) focused on employing kernels to generalize the model to non-linear transformation. DCCA (Andrew et al. (2013)) introduced deep learning to CCA and replaced the core matrix transformation with a multi-layered neural network, and DTCCA (Wong et al. (2021)) further extended this to more than two views. With progress in variational neural networks (Kingma & Welling (2013)), variational CCA (VCCA) (Wang et al. (2016)) re-formulated CCA under a graph model and proposed variational CCA using a generative framework. VCCA assumed multimodal data were generated from a shared latent representation and solved it under the variational autoencoder framework. Similarly, deep probabilistic CCA (DPCCA) (Karami & Schuurmans (2021)) took a probabilistic interpretation of CCA and decomposed the multimodal features into a shared distribution and a modality-specific distribution. We note VCCA and DPCCA are different from the original CCA and DCCA as they combined all modalties into a joint representation instead of learning a unique transformation for each modality. Our proposed adCCA is different from these approaches in that our model is strictly formulated within the classical CCA framework.

The generative adversarial network (GAN) (Goodfellow et al. (2014)) framework was originally proposed to learn generative processes underlying a data distribution. The adversarial autoencoder (Makhzani et al. (2015)) was the first work to use adversarial loss to match autoencoder latent representation to any prior distribution. A combination of CCA and GAN was used for ACCA (Dutton (2020)), which can be viewed as an extension of the variational CCA. Instead of using a Kullback–Leibler (KL) divergence to constrain the prior and posterior distributions in latent space, ACCA used an adversarial loss to match the latent representation to arbitrary priors. DACCA (Fan et al. (2020)) concatenates multiview generation and deep CCA for data generation and representation learning. Most adversarial learning frameworks focus on constraining the distributions between generated and real samples, rather than between latent representations from different modalities. Our proposed adCCA is different from existing approaches in that adCCA directly uses the adversarial loss to constrain the latent distributions from multimodal features. We summarize the major differencse in representation learning and training for CCA variants in Appendix Table F.6.

## B  MATHEMATICAL FORMULATIONS

### B.1  OPTIMIZATION OF GENERAL CCA FRAMEWORK

$\mathbf{Z}_x \in \mathbb{R}^{n \times k}$ and $\mathbf{Z}_y \in \mathbb{R}^{n \times k}$ are mean-centered matrix form representations from two modalities with functions $f_{\mathbf{w}_x}^{(x)}(\mathbf{X})$ and $f_{\mathbf{w}_y}^{(y)}(\mathbf{Y})$ parameterized by $\mathbf{w}_x$ and $\mathbf{w}_y$, where $\mathbf{X} \in \mathbb{R}^{n \times p}$ and $\mathbf{Y} \in \mathbb{R}^{n \times q}$. The correlation can be formulated as a matrix trace norm:

$$
\begin{aligned}
corr(\mathbf{Z}_x, \mathbf{Z}_y) &= \|\mathbf{S}\|_{trace} \\
\mathbf{S} &= \mathbf{\Sigma}_{xx}^{-1/2} \mathbf{\Sigma}_{xy} \mathbf{\Sigma}_{yy}^{-1/2},
\end{aligned}
\tag{B.1}
$$

where $\mathbf{\Sigma}_{xx} = \frac{1}{n-1}\mathbf{Z}_x^T \mathbf{Z}_x$, $\mathbf{\Sigma}_{yy} = \frac{1}{n-1}\mathbf{Z}_y^T \mathbf{Z}_y$, and $\mathbf{\Sigma}_{\mathbf{xy}} = \frac{1}{n-1}\mathbf{Z}_x^T \mathbf{Z}_y$. Based on Andrew et al. (2013), the optimization of $\mathbf{w}_x$ and $\mathbf{w}_y$ requires the gradient of correlation to be calculated:

$$
\begin{aligned}
\frac{\partial corr(\mathbf{Z}_x, \mathbf{Z}_y)}{\partial \mathbf{w}_x} &= \frac{\partial corr(\mathbf{Z}_x, \mathbf{Z}_y)}{\partial \mathbf{Z}_x} \cdot \frac{\partial \mathbf{Z}_x}{\partial \mathbf{w}_x} \\
&= \frac{1}{n-1}(2M_{xx}\mathbf{Z}_x + M_{xy}\mathbf{Z}_y) \cdot \frac{\partial f_{\mathbf{w}_x}^{(x)}}{\partial \mathbf{w}_x},
\end{aligned}
\tag{B.2}
$$

where $M_{xy} = \mathbf{\Sigma}_{xx}^{-1/2}\mathbf{U}\mathbf{V}^T\mathbf{\Sigma}_{yy}^{-1/2}$ and $M_{xx} = -\frac{1}{2}\mathbf{\Sigma}_{xx}^{-1/2}\mathbf{U}\mathbf{D}\mathbf{U}^T\mathbf{\Sigma}_{yy}^{-1/2}$. $\mathbf{U}$ and $\mathbf{V}$ are from the *Singular Value Decomposition* (SVD) of $\mathbf{S}$ by $\mathbf{S} = \mathbf{U}\mathbf{D}\mathbf{V}^T$. The term $\frac{\partial corr(\mathbf{Z}_x, \mathbf{Z}_y)}{\partial \mathbf{w}_y}$ can be formulated similarly.

The calculation in Eq. B.2 requires the SVD decomposition on all samples for every training iteration. Therefore, it is not tractable for standard mini-batch training.

## B.2 PROOF OF THEOREM 1

*Proof.* $\mathbf{x} \in \mathbb{R}^p$ and $\mathbf{y} \in \mathbb{R}^q$ represent a pair of random variables with $p$ and $q$ dimensions in two modalities. $z_x = f_1(\mathbf{x}) \in \mathbb{R}$ and $z_y = f_2(\mathbf{y}) \in \mathbb{R}$ maps modality features to a pair of canonical variables with transformations $f_1$ and $f_2$. With that, a general optimization target for CCA can be formulated as:

$$
\begin{aligned}
f_1^*, f_2^* &= \arg\max_{f_1, f_2} corr(z_x, z_y) \\
&= \arg\max_{f_1, f_2} \frac{\mathbb{E}[(z_x - \mu_x)(z_y - \mu_y)]}{\sigma_x \sigma_y} \\
&= \arg\max_{f_1, f_2} \frac{\mathbb{E}(z_x z_y) - \mu_x \mu_y}{\sigma_x \sigma_y},
\end{aligned}
\tag{B.3}
$$

where $\mu_x$ and $\sigma_x$ are mean and standard deviation for $z_x$ (*Resp.* $\mu_y$ and $\sigma_y$ for $z_y$). Here, $z_x$ and $z_y$ are assumed to follow the same distribution $\pi$ with shared mean and standard deviation $\mu$ and $\sigma$ and rewrite the target to:

$$
\begin{aligned}
f_1^*, f_2^* &= \arg\max_{f_1, f_2} \frac{\mathbb{E}(z_x z_y) - \mu_x \mu_y}{\sigma_x \sigma_y}, \\
&= \arg\max_{f_1, f_2} \frac{\mathbb{E}(z_x z_y) - \mu^2}{\sigma^2}, \\
&\underset{\mu, \sigma \in C}{=} \arg\max_{f_1, f_2} \mathbb{E}(z_x z_y),
\end{aligned}
\tag{B.4}
$$

where $C$ is the constant number set.

$\square$

## B.3 PROOF OF COROLLARY 1

*Proof.* $\mathbf{x} \in \mathbb{R}^p$ and $\mathbf{y} \in \mathbb{R}^q$ are random variables with $p$ and $q$ dimensions from two modalities. Two canonical representation vectors $\mathbf{z}_x$ and $\mathbf{z}_y$ with $k$ dimensions are learned from two autoencoders and constrained by the correct reconstruction of original features. The $j^{\text{th}}$ entries of $\mathbf{z}_x$ and $\mathbf{z}_y$ follow the latent distribution with fixed mean $\mu$ and variance $\sigma^2$. Following the Theorem 1, the learning target for CCA with multiple canonical variables can be generalized as:

$$
\begin{aligned}
f_1^*, f_2^* &= \arg\max_{f_1, f_2} \sum_{j=1}^{k} corr(\mathbf{z}_x^{(j)} \mathbf{z}_y^{(j)}) \\
&= \arg\max_{f_1, f_2} \sum_{j=1}^{k} \mathbb{E}(\mathbf{z}_x^{(j)} \mathbf{z}_y^{(j)}) \\
&= \arg\max_{f_1, f_2} \sum_{j=1}^{k} \frac{1}{n} \sum_{i=1}^{n} z_x^{(i,j)} z_y^{(i,j)} \\
&= \arg\max_{f_1, f_2} \frac{1}{n} \sum_{i=1}^{n} \left[ \sum_{j=1}^{k} z_x^{(i,j)} z_y^{(i,j)} \right] \\
&= \arg\max_{f_1, f_2} \frac{1}{n} \sum_{i=1}^{n} z_x^{(i,:)T} z_y^{(i,:)} \\
&= \arg\max_{f_1, f_2} \mathbb{E}[\mathbf{z}_x^{T} \mathbf{z}_y],
\end{aligned}
\tag{B.5}
$$

where $\mathbf{z}_x^{(j)}$ is the $j^{\text{th}}$ entry of the vector; $z_x^{(i,j)}$ is the $j^{\text{th}}$ entry of the representation learned from sample $i$; $z_x^{(i,:)}$ is the whole representation vector from sample $i$ (*Resp.* $\mathbf{z}_y^{(j)}$, $z_y^{(i,j)}$ and $z_y^{(i,:)}$ for modality $y$). We note compared with the SVD-based solution in Eq. B.2, different dimensions in the latent representations are no long orthogonal to each other. $\square$

## C  BACKGROUND

### C.1  AUTOENCODERS

Autoencoders are a general unsupervised representation learning framework. Given a sample feature vector $\mathbf{x} \in \mathbb{R}^p$, an autoencoder tries to learn a compact feature $\mathbf{z} \in \mathbb{R}^q$ from an encoder network that can be faithfully reconstructed to the original feature by a decoder network. The whole framework is optimized by minimizing the reconstruction error:

$$
\min_{\varphi, \theta} \mathbb{E}_{\mathbf{x} \sim p_x} \| r_\varphi[g_\theta(\mathbf{x})] - \mathbf{x} \|_2^2,
\tag{C.1}
$$

where $g_\theta$ and $r_\varphi$ are encoder and decoder networks parameterized by $\theta$ and $\varphi$, respectively; $p_x$ is the distribution of input. The representation is the output from the encoder $\mathbf{z} = g_\theta(\mathbf{x})$.

### C.2  WASSERSTEIN GAN

The Wasserstein GAN is an improved GAN framework. It modifies the core classification network by using the Wasserstein distance to measure the distribution difference between the generated and realistic samples. Given a sample $\mathbf{x}$ and a prior distribution $\mathbf{z} \sim p_z$, the Wasserstein GAN is learned via a two-step minmax optimization:

$$
\min_{\theta} \max_{w} \mathbb{E}_{\mathbf{x} \sim p_x} f_w(\mathbf{x}) - \mathbb{E}_{\mathbf{z} \sim p_z} f_w[g_\theta(\mathbf{z})],
\tag{C.2}
$$

where $g_\theta$ and $f_w$ are generator and discriminator with parameter $\theta$ and $w$; $p_x$ is the actual data distribution and $p_z$ is the prior distribution to generate new samples.

# D SUPPLEMENTARY FIGURES

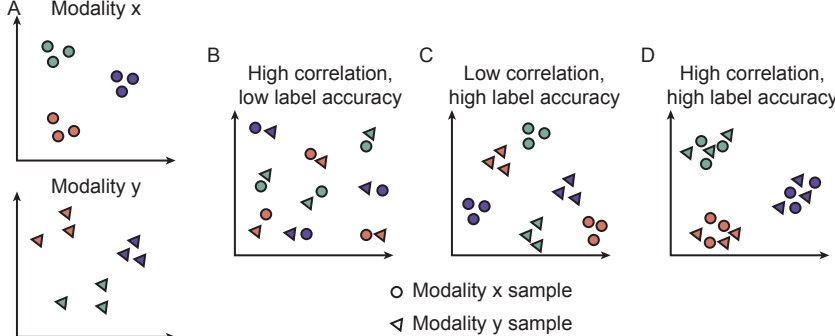

Figure D.1: Cartoon illustration of different possible outcomes from canonical representation learning of two modalities. (A) Sample raw features from two modalities visualized in 2D. Color indicates the class labels. (B-D) Three scenarios of canonical representations learned from two modalities. (B) 2D embedding of canonical representations with high cross-modality correlation but low class information. (C) 2D embedding of canonical representations with well separation between classes but low correlation between modalities. (D) Ideal case that canonical representations from two modalities and same classes are well mixed, meanwhile different classes are well separated.

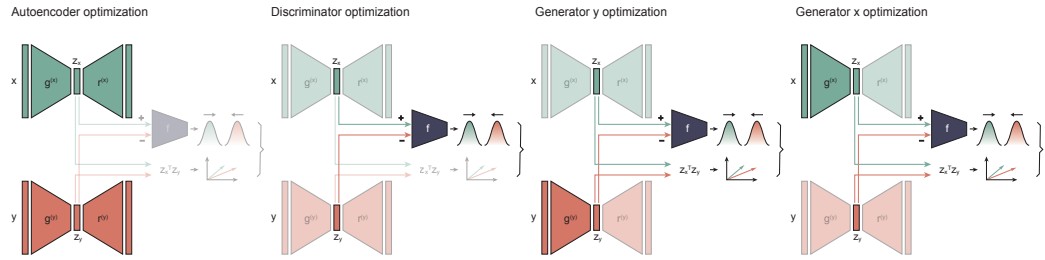

Figure D.2: Optimization steps for adCCA. Transparent structures represent modules not involved in the optimization step.

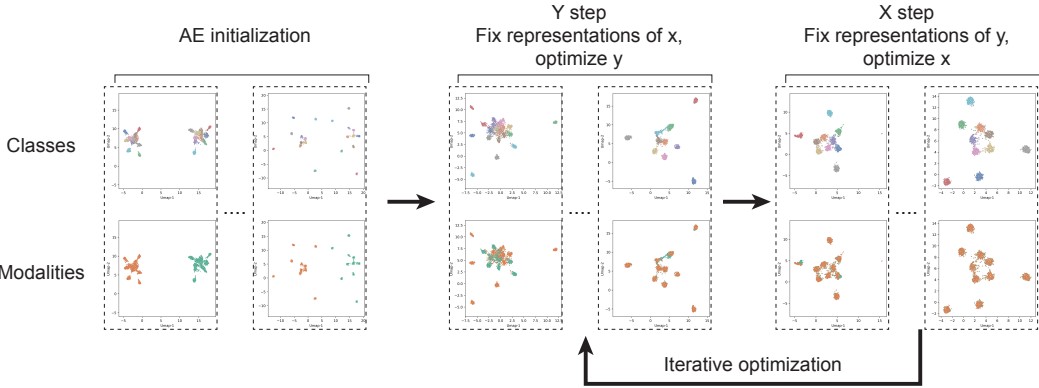

Figure D.3: Visualization of how two modalities were aligned in the latent space during adCCA optimization. Latent representations were extracted during training and were mapped to 2D space for visualization using Umap. Samples were annotated with classes (top row) or modalities (bottom row).

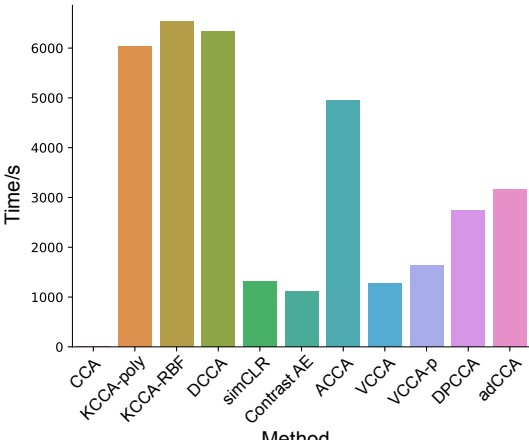

Figure D.4: Runtime of comparing approaches on MNIST dataset.

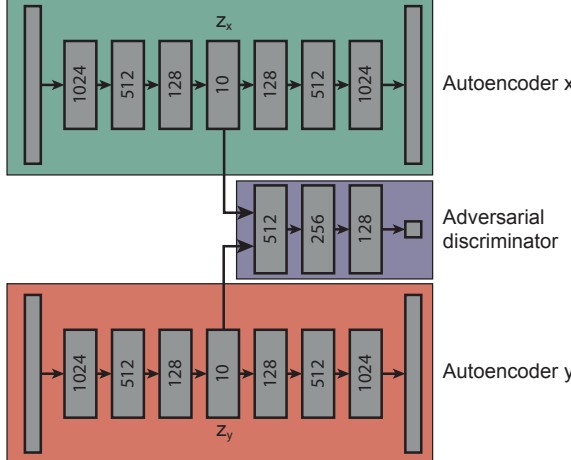

Figure D.5: Network structure in adCCA experiments. Each block includes 1D batch normalization and ReLU activation function.

# E EXPERIMENT DETAILS

## E.1 COMPARISON OF APPROACHES

Here we set the latent dimension to 10 for all approaches. For deep-learning based methods, we used the same encoding network $[1024, 512, 256, 10]$ to learn the latent representation.

**CCA:** we used the multiview CCA function in the `mvlearn` python package [1] (version 0.5.0) with regulation paramter `regs=0.5`.

**KCCA:** kernel CCA is implemented from the in `KMCCA` function in `mvlearn` python package. For Radial Basis Function (a.k.a) kernel, we set $\gamma = 1$, $regs = 0.01$, $sval - thresh = 1e - 5$. For polynomial kernel, we set kernel parameter as $degree = 2$, $coef0 = 0.1$ and $regs = 0.01$.

**DCCA:** we used the DCCA function in `mvlearn` and set the training epoch to 500. Default values are used for all other parameters.

---

[1]https://mvlearn.github.io

**simCLR:** we made use of the `NT_Xent` simCLR implementation[2] and set the training epoch to 500. The learning step is chosen by grid search from $[1e-2, 1e-3, 1e-4]$.

**Contrast AE:** the implementation is based on the adCCA framework. The adversarial learning structure is removed and only the similarity constraint is used in learning. The parameter settings and training strategy are the same as in adCCA.

**ACCA:** we used the implementation from the authors[3]. The latent representation is generated from both modalities ($q(z|x, y)$ mode by setting $num\_z = 2$). The $L_2$ reconstruction loss is used.

**VCCA** and **VCCA-p:** we used the implementation from ACCA. We followed the original VCCA framework and assume the joint latent representation is generated from single view.

**DPCCA:** we modified the model from the original paper[4]. Parameters were selected from a grid search of learning steps $[1e-2, 1e-3, 1e-4]$ and epochs $[500, 1000, 2000]$. $\lambda$ was set to 1, but we empirically determined it has little effect on the final results.

**adCCA:** the model is implemented with the `pyTorch` package. The network structure used in the experiment is presented in Apendix Fig. D.5. Each layer includes a `BatchNorm1d` and a `ReLU` activation function. The Adam optimizer was used for both the AE and GAN networks. $\beta$ (coefficients for the running averages of the gradient and its square) was selected from a grid search, giving $[0.9, 0.99]$ for the discriminator and $[0.5, 0.9]$ for the generator and AE. The hyperparameter $\lambda$ balancing the inner product term and Wasserstein distance was set to 1. Each training step includes 500 epochs. The learning rate was set to $1e-3$. In each iteration, the discriminator optimizer was repeated twice and the generator optimizer repeated 5 times for better convergence in each step. In the GAN optimization, we used the wGAN with gradient penalty [5]. Source code and demonstration to run the code are provided in supplementary materials.

## E.2 SIMULATION DESIGN

We first pre-determined the $k$ classes existing in the data. To get raw features $\mathbf{X} \in \mathbb{R}^{n \times p}$ for modality $x$, we use a multivariate normal distribution with randomly determined mean to generate the latent code $\mathbf{Z}_x \in \mathbb{R}^{n \times d}$. Next, we projected the latent code to a new space with a multi-layer neural network:

$$\begin{aligned}
\mathbf{X}^{(1)} &= \sigma(\mathbf{Z}_x \mathbf{W}_1^{(x)}) \\
\mathbf{X}^{(2)} &= \sigma(\mathbf{X}^{(1)} \mathbf{W}_2^{(x)}),
\end{aligned} \tag{E.1}$$

where $\sigma(\cdot)$ is the sigmoid function; $\mathbf{W}_1^{(x)} \in \mathbb{R}^{d \times m}$ and $\mathbf{W}_2^{(x)} \in \mathbb{R}^{m \times p}$ are random projection matrices. Finally, we added random noise and random feature dropouts to obtain the final feature:

$$\mathbf{X} = \delta^{(x)}(\mathbf{X}^{(2)} + \varepsilon; \xi), \tag{E.2}$$

where $\delta^{(x)}(\cdot; \xi)$ indicates the random feature sampling at a probability $\xi \in [0, 1]$; $\varepsilon$ is a Gaussian noise matrix with the same shape as $\mathbf{X}^{(2)}$. The $\mathbf{X}$ is the final input feature for modality $x$. Similarly, we obtained input feature $\mathbf{Y} \in \mathbb{R}^{n \times q}$ for modality $x$ for modality $y$ using the same framework with different random projection matrices and drop-out function.

In the experiment, we set $k = 50$, $p = q = 1,000$, and noise variances to 0.1 and 0.9 for $x$ and $y$, respectively. Further, 90% and 10% of features are randomly missing for these two modalities.

In the experiment of feature transformations (Section 3.1.2), for the feature distribution transformation, we linearly shifted features with a minimal value 0 then employed two sequential $\log 1p$

---

[2]`https://github.com/Spijkervet/SimCLR/blob/master/simclr`
[3]`https://github.com/bcdutton/AdversarialCanonicalCorrelationAnalysis`
[4]`https://github.com/Karami-m/Deep-Probabilistic-Multi-View`
[5]`https://github.com/eriklindernoren/PyTorch-GAN/blob/master/`
`implementations/wgan_gp/wgan_gp.py`

transformations; for the feature encoding transformation, we used local sensitive hashing (LSH) (Datar et al. (2004)) implemented from GitHub[6] to encode features into $1,000$ bits.

### E.3 EVALUATION METRIC

**Entropy of mixing:** we put latent representations from $x$ and $y$ together $[\mathbf{z}_x^T; \mathbf{z}_y^T]^T$. In the combined data, we identify its 100 nearest neighbors for each sample and calculate the proportion of neighbors from the same or different modality. The entropy of mixing is given by:

$$-[p \log p + (1-p) \log(1-p)], \tag{E.3}$$

where $p$ is the proportion of neighbors belonging to modality $x$.

**Classification analysis:** we employed a 5-fold classification accuracy evaluation. In each data split, we fit a linear support vector machine (SVM) or $k-$nearest-neighbor classifier ($k$NN) on training set (80%) and calculated classification accuracies on the test set (20%). We used implementations of both classifiers from `sklearn`. For $k$NN, we considered 10 nearest neighbors to build classifier.

**Silhouette score:** we use the implementation from `sklearn`. Given the true sample classes, $d_a$ is the average distance between samples in the same class, $d_b$ is the average distance between one sample and its nearest samples from different classes. The score is defined as:

$$\frac{d_b - d_a}{\max(d_a, d_b)}, \tag{E.4}$$

a positive value indicates the inner-class-distance is smaller than between-class-distance.

### E.4 MNIST EXPERIMENTS

We applied rotation (range $[-10°, 10°]$), translation (range [0,0.1]), scaling (range [0.9,1.1]), shearing ([0,15]) and random pixel corruption (10%) to digit images. Each view was obtained through this transformation sequence with randomly chosen transformation parameters. After the transformation, two views were input to approaches, and the results were compared with the original digit labels.

In the experiments with increased corruption levels, we further increased the range of each transform to 2 fold and 3 fold (*e.g.*, the rotation angle range was broaded to $2*[-10°, 10°]$ and $3*[-10°, 10°]$) and then tested accuracy.

### E.5 CITESEQ EXPERIMENTS

The dataset was downloaded from a previous publication[7]. We used the DLN111-D1 batch, which includes 9,264 cells from spleen and lymph node in total. Each cell includes genes ($> 20,000$ dimensions) and surface proteins (112 dimensions). For the gene modality, we selected the top 2,000 variable genes as input; for the protein modality, we input all proteins. Cells are classified to 5 major cell types and 35 cell subtypes.

## F SUPPLEMENTARY TABLES

---

[6]`http://ethen8181.github.io/machine-learning/recsys/content_based/lsh_text.html`

[7]`https://github.com/YosefLab/totalVI_reproducibility/tree/master/data`

Table F.1: Model structure analysis of adCCA. Note: removing Wasserstein distance term is equivalent to Contrast AE approach.

| Method | Correlation | SVM |
|---|---|---|
| adCCA | $0.987 \pm 0.011$ | $0.991 \pm 0.003$ |
| Remove AE initialization | $0.978 \pm 0.006$ | $0.310 \pm 0.106$ |
| Remove Wasserstein distance | $-0.074 \pm 0.396$ | $0.294 \pm 0.032$ |
| Remove inner product | $0.282 \pm 0.014$ | $0.134 \pm 0.023$ |
| Fix decoders | $0.772 \pm 0.045$ | $0.327 \pm 0.090$ |

Table F.2: Consistency evaluation of log-transformed feature.

| Method | Correlation | Entropy of Mixing |
|---|---|---|
| CCA | $0.972 \pm 0.002$ | $0.692 \pm 0.002$ |
| KCCA-poly | $0.558 \pm 0.029$ | $0.403 \pm 0.198$ |
| KCCA-RBF | $0.993 \pm 0.001$ | $\mathbf{0.693 \pm 0.000}$ |
| DCCA | $0.982 \pm 0.010$ | $0.690 \pm 0.005$ |
| simCLR | $0.323 \pm 0.131$ | $0.686 \pm 0.012$ |
| Contrast AE | NA | NA |
| adCCA | $\mathbf{0.998 \pm 0.002}$ | $0.692 \pm 0.002$ |

Table F.3: Class of evaluation of log-transformed feature.

| Method | ARI | Silhouette | SVM | $k$NN |
|---|---|---|---|---|
| CCA | 0.032 | -0.022 | $0.118 \pm 0.026$ | $0.365 \pm 0.020$ |
| KCCA-poly | 0.563 | 0.152 | $0.124 \pm 0.040$ | $0.721 \pm 0.265$ |
| KCCA-RBF | 0.000 | -0.395 | $0.111 \pm 0.010$ | $0.967 \pm 0.004$ |
| DCCA | 0.515 | 0.148 | $0.717 \pm 0.013$ | $0.710 \pm 0.013$ |
| simCLR | 0.021 | -0.038 | $0.223 \pm 0.068$ | $0.151 \pm 0.033$ |
| Contrast AE | NA | NA | NA | NA |
| ACCA | 0.014 | -0.035 | $0.128 \pm 0.025$ | $0.155 \pm 0.005$ |
| VCCA | 0.000 | -0.024 | $0.112 \pm 0.017$ | $0.105 \pm 0.007$ |
| VCCA-p | 0.013 | -0.063 | $0.123 \pm 0.017$ | $0.131 \pm 0.012$ |
| DPCCA | 0.147 | -0.047 | $0.354 \pm 0.018$ | $0.333 \pm 0.017$ |
| adCCA | **0.651** | **0.265** | $\mathbf{0.995 \pm 0.002}$ | $\mathbf{0.993 \pm 0.002}$ |

Table F.4: Consistency evaluation of binary encoded features.

| Method | Correlation | Entropy of Mixing |
|---|---|---|
| CCA | $0.972 \pm 0.003$ | $0.692 \pm 0.002$ |
| KCCA-poly | $0.762 \pm 0.008$ | $0.517 \pm 0.140$ |
| KCCA-RBF | $\mathbf{0.998 \pm 0.001}$ | $\mathbf{0.693 \pm 0.000}$ |
| DCCA | $0.993 \pm 0.004$ | $0.691 \pm 0.003$ |
| simCLR | $0.274 \pm 0.121$ | $0.688 \pm 0.006$ |
| Contrast AE | NA | NA |
| adCCA | $0.995 \pm 0.005$ | $0.692 \pm 0.002$ |

Table F.5: Class of evaluation of binary encoded features.

| Method | ARI | Silhouette | SVM | $k$NN |
|---|---|---|---|---|
| CCA | 0.013 | -0.029 | $0.111 \pm 0.011$ | $0.297 \pm 0.003$ |
| KCCA-poly | 0.050 | -0.023 | $0.117 \pm 0.025$ | $0.276 \pm 0.085$ |
| KCCA-RBF | 0.002 | -0.020 | $0.107 \pm 0.002$ | $0.178 \pm 0.021$ |
| DCCA | 0.332 | 0.052 | $0.611 \pm 0.009$ | $0.585 \pm 0.011$ |
| simCLR | 0.006 | -0.033 | $0.151 \pm 0.022$ | $0.117 \pm 0.007$ |
| Contrast AE | NA | NA | NA | NA |
| ACCA | 0.475 | **0.116** | $0.493 \pm 0.028$ | $0.701 \pm 0.016$ |
| VCCA | 0.427 | 0.038 | $0.928 \pm 0.017$ | $0.817 \pm 0.018$ |
| VCCA-pr | 0.002 | -0.043 | $0.130 \pm 0.017$ | $0.123 \pm 0.014$ |
| DPCCA | 0.003 | -0.028 | $0.119 \pm 0.021$ | $0.102 \pm 0.011$ |
| adCCA | **0.530** | 0.100 | $\mathbf{0.943 \pm 0.004}$ | $\mathbf{0.934 \pm 0.010}$ |

Table F.6: A summary of representation and training for CCA variants.

| Method | Representation type | Training |
|---|---|---|
| CCA | Modality-specific representation | Large/full batch |
| KCCA-poly | Modality-specific representation | Large/full batch |
| KCCA-RBF | Modality-specific representation | Large/full batch |
| DCCA | Modality-specific representation | Large/full batch |
| ACCA | Single joint representations | Mini-batch |
| VCCA | Single joint representations | Mini-batch |
| VCCA-p | Single joint representations | Mini-batch |
| DPCCA | Single joint representations | Mini-batch |
| adCCA | Modality-specific representation | Mini-batch |

