# OpenReview forum: "Adversarial Representation Learning for Canonical Correlation Analysis"
_ICLR.cc/2023/Conference — Submitted to ICLR 2023_

### Official Review · Reviewer_fcpJ · 2022-10-24

**Confidence:** 4
**Correctness:** 2
**Technical Novelty And Significance:** 2
**Empirical Novelty And Significance:** 2
**Recommendation:** 3

**Clarity, Quality, Novelty And Reproducibility:**

Clarity: The current version of paper is not clearly presented. The paper can be further polished to make it easier to follow.
Quality: The technique parts, for instance theorems, need to be improved, otherwise I don't think the paper is ready to publish.
Novelty: I think the novelty is very limited.
Reproducibility: No enough datapoint as I don't try their code yet.




**Strength And Weaknesses:**

Strengths:
- They conducted extensive experiments to demonstrate the effectiveness of their method

Main concerns:
- The paper is hard to follow due to the incoherence and imprecision in their writing. For instance, the second paragraph in introduction section "it is temping ...". It is unclear to me what does it refer to in this sentence. the second paragraph on page 2, "This avoids having to balance the two modalities." I don't quite get what messages the authors try to convey via this sentence.
- All theorems were built on a faithless assumption. They assume "latent representations from modality x and y follow the same distribution $\pi$ with the mean $\mu$ and variance $\sigma^2$" which is weird to me. In practice, the distribution discrepancy is commonly occurred among data that come from different sources/views/modalities. At least, at the initial stage I don't think the representation distribution obey their assumption. However, they derived their theorems from their fundamental assumption which is untrustworthy.
- The authors emphasized that their method is able to learn maximally correlated representations without comprising structure in each modalities. I was wondering how do the authors prove the structures among modalities were not comprised. How should I understand the correlation of representations is maximized while the structure are not deformed? I would very much appreciate the authors if they can help me understand this.
- I don't think it is fair to claim resolving adCCA with mini-batch optimization as one major contribution of this work since  DCCA (Wang et al.) already confirmed stochastic mini-batch optimization works for DCCA based methods.
- I would expect the authors compare their method with [1,2] from the methodology perspective.
[1]. Benjamin Dutton, Adversarial Canonical Correlation Analysis, 2020
[2] Fan et al., Deep Adversarial Canonical Correlation Analysis, 2020

**Summary Of The Paper:**

This work derived a new deep adversary CCA method, which is called adCCA, from DCCA framework. The authors frame CCA problem under the assumption that the representations of different modalities obey an identical distribution in a common latent subspace. Compared to other DNN-based CCA methods, the authors claim 1) their approach is able to learn highly correlated representations across modalities, without deforming original structure of each modal. 2) they simplified analytical solution of CCA problem with their identical distribution assumption. 3) extensive experiments valid their method and demonstrate the robustness of their model.

**Summary Of The Review:**

The current version of paper is not ready to publish since there are some concerns need to be addressed.

---

> ### Author Response · Authors · 2022-11-14
> **Improve clarity and accuracy of writing, experiment and theorem**
>
> We appreciate the reviewer’s constructive comments on our work. We particularly took to heart the need to provide more clarity on motivation and methods. We hope the revised text, figures and experiments address these concerns. Please let us know if the reviewer has further questions.
> ## Reply to point 1
> We agree with your points on clarity and have revised these sections.
>
> ## Reply to point 2
> We apologize that we did not clearly explain the connection between the mathematical assumption and the implication.
>
> Theory: We make the mathematically simplifying assumption that the different modality distributions are identical in latent space. This allows us to derive an optimization target for CCA.
>
> Implementation: The theorical assumption of identical latent distributions is formulated as a penalty function (using the Wasserstein distance) for adCCA. Of course, upon initialization, there is no reason to expect that the latent representations of the modalities are the same. However, during optimization, this penalty function brings the two distributions into alignment. As the latent space representations converge (distribution-wise), maximizing the optimization target will also lead to highly correlated latent representations (sample-wise) for the two modalities. The visualization of the latent representations during training steps illustrates that this indeed happens (Appendix Figure D3).
>
> We have revised the manuscript (Introduction) to make the connection between theory and implementation clearer, as well as added a Lagrangian formulation of the adCCA learning target (Eq. 8) that brings these ideas together.
>
> ## Reply to point 3
>
> In the manuscript, we used “structure” to describe labeled datapoints that are geometrically grouped in feature space within each modality. Our assessment of whether structures are or are not compromised by adCCA is based on the ability of supervised or unsupervised algorithms to correctly predict class labels for each datapoint (Table 2).  These assessments are performed in latent space, which are deformed transformation of the raw features.
>
> To avoid misunderstanding, we replaced the terminology “structure” with “label/class”.
>
> ## Reply to point 4
>
> We greatly appreciate the reviewer pointing us to this work. We went through the work carefully. Based on the conclusion in this work, the authors state that “DCCA can still be optimized efficiently as long as the gradient is estimated using a sufficiently large minibatch”. “Sufficiently large” is somewhat vague, but we take it to mean that standard min-batch formulations are not adequate. Our approach does not require this qualification.
> We have revised the abstract to state that other approaches require “sufficiently large mini-batches.”
>
> ## Reply to point 5
>
> The first work, ACCA, is a variational multiview autoencoder structure, which aims to learn a joint representation from multi-modal data. The adversarial network in ACCA is used to force the joint representation to follow a prior distribution (e.g., standard Gaussian distribution). Their adversarial network is not designed to guide CCA learning. As comparison, the adversarial structure in adCCA is designed to simultaneously minimize distribution-wise difference in latent space representations while maximizing sample-wise correlation. The second work is focused on using a GAN for better sample generation of multi-modal data.
> The first work ACCA has been benchmarked in the original and current submissions. The second work has a different research focus and is not relevant for comparison to adCCA.

---

> > ### Comment · Reviewer_fcpJ · 2022-11-23
> > **Thanks for authors' response**
> >
> > Thank the authors for addressing most of my concerns. I will keep my scores and recommend to further polish your work.

---

### Official Review · Reviewer_NwB7 · 2022-10-24

**Confidence:** 4
**Correctness:** 4
**Technical Novelty And Significance:** 4
**Empirical Novelty And Significance:** 4
**Recommendation:** 5

**Clarity, Quality, Novelty And Reproducibility:**

The proposed method is interesting, but the current submission has several issues that need to be addressed. Some details are missing. I write my main concerns and comments in the box of [Strength And Weaknesses].


**Strength And Weaknesses:**

The proposed method seems interesting. The experiments are sufficient to show the effectiveness of the proposed adCCA. I have some concerns listed as follows:

1. The preliminary results show that latent features $z_x$ and $z_y$ have the same latent distribution, with a mean of $mu$ and a variance of $sigma2$. And section 2.2 presents this shared latent distribution is achieved by using batch normalization. This is confusing information. As illustrated in Figure 1, adCCA employs two distinct neural networks, $g(x)$ and $g(y)$, to extract the corresponding latent features. However, it's difficult to understand how to use the BN techniques to constrain the two modality features to follow the same distribution.
2. On page 5, the authors said that ''This permits the use of the modality specific structures and removes the requirement of modality balance.'' Although this sentence is easy to follow, this statements are not supported by experiments or conclusions from previous works. I think it's better to clarify these statements (maybe from relevant literatures). Or this statements seem to be the authors' hypothesis.
3. The descriptions of the neural networks through the whole paper are missing, which is very important for Reproducibility.
4. According to the Figure 5 and Table 3, we find that higher correlation may not lead to better class prediction. What is the cause of this phenomenon? Is there a trade-off problem between correlation and accuracy? Moreover, we also find that most of the joint representation learning methods achieve worse results. Is this a problem inherent in this type of approach? Or is this problem due to the asymmetric information between different modalities? I think the authors should give more analysis in the experiments.
5.In Algorithm 1, the initialization of $z_x$ and $z_y$ is link to a pre-trained model. So, how much does this initialization process affect the result? For example, do we remove it? Because in the next part, we also update the parameter of the AE network. Moreover, after we achieve two pre-trained AEs, shall we fix the parameter $\Phi$ during the second iterative optimization?


**Summary Of The Paper:**

This paper studies the canonical correlation analysis under an adversarial learning framework. They propose an effective adCCA that can learn better representations. The experiments show the proposed method is competitive compared to many recent deep CCA.

**Summary Of The Review:**

This paper proposed a deep CCA model under adversarial learning framework. The proposed method is novel and the experiments are sufficient.

---

> ### Author Response · Authors · 2022-11-14
> **Clarification on model structure, experiments, and additional results on structure analysis**
>
> We appreciate the reviewer’s careful read of our manuscript. The comments were very constructive and have improved the revised manuscript. Please let us know if you have further questions.
>
> ## Reply to Comment 1
>
> The reviewer is correct that BN can only standardize the data to have the same mean and variance and cannot constrain the two modality features to follow the same distribution. In our model, the two latent representations from each modality are encouraged to follow the same distribution by minimizing their Wasserstein distance.
>
> To avoid this misunderstanding, we updated Figure 1 to highlight the function of the Wasserstein distance and added Eqs. 5, 6, 7 for a detailed description of how to measure and minimize the distribution difference.
>
> ## Reply to Comment 2
>
> Thank you for this suggestion. We revised the sentence to now point the reader to the experiment results in Section 3.1.2, where we performed data distribution transformations and showed robust performance from adCCA. The sentence now reads: “This enables adCCA to work with raw input feature data from different modalities and different distributions (see section 3.1.2).”
>
> ## Reply to Comment 3
>
> Yes, this is important! Thank you. We added Appendix Fig. D.5 to provide the network structure used in the experiment and revised Appendix E.1 to give a more complete description of parameters used.
>
> ## Reply to Comment 4
>
> Thank you for the question about correlation vs class prediction accuracy. To address your point, we revised the introduction and added Fig 1 to illustrate that correlation and accuracy are two separate measures for evaluation and that one may not imply the other. We have also revised the end of the experimental section 3.1.1 to clarify this point.
>
> You are correct. Many joint representation learning methods achieve worse results, partially due to the asymmetric information between two modalities. We performed log transformation to create asymmetric feature distributions for two (originally similar) modalities (Fig. 4A bottom). This transformation had dramatic impact to the performance of the joint representation learning methods, suggesting that they are not stable to these types of changes that imbalance the data.
>
> ## Reply to Comment 5
>
> Thanks for the comment. In the revision, we tested the effect of autoencoder initialization by removing this step and re-ran the model. From the result, initialization had small effects on the correlation between representations learned from two modalities (because the strong constraint from Wasserstein distance and inner product terms) but strong effects on class prediction (due to the insufficient training of the AE).
>
> Further, we tested fixing the decoder parameter after the AE initialization and only optimized the encoder in the second step. Because the decoder parameters do not get optimized, this further hindered the latent representation from reaching a better balance point between cross-modality consistency and single-modality reconstruction. Therefore, we observed both decreases in correlation and classification metrics.
>
> These studies are now included as Appendix Table F.1.

---

### Official Review · Reviewer_yzLx · 2022-10-30

**Confidence:** 3
**Correctness:** 2
**Technical Novelty And Significance:** 3
**Empirical Novelty And Significance:** 3
**Recommendation:** 3

**Clarity, Quality, Novelty And Reproducibility:**

In light of all of my questions above, I have to say clarity is quite low. Quality is uncertain. I am not familiar enough with recent work to be very confident about novelty. I do know that using separate encoders for each modality, including autoencoders, has been tried. I also know there have been other methods proposed for training DCCA-style models without having to use full-batch optimization as in the original paper. If indeed the approach is novel, it should probably be compared with some of these existing techniques in terms of training time. Since the approach combines several things (autoencoding initial layers, penalizing Wasserstein distance, using dot product objective that can be decomposed over examples) it would also be good to have ablation experiments to determine which of these components are necessary for the claimed performance.

**Strength And Weaknesses:**

The paper employs some interesting techniques and could be of value, but a few points were not clear to me.

1) The proposed technique is claimed to have the advantage of not needing to "balance the two modalities". What is meant by that? How do existing methods for learning correlated representations suffer from "unbalanced modalities"? Using some modality-specific transformation certainly sounds reasonable, independent of concerns about not being "balanced".
2) The derivation of Theorem 1 and (especially) corollary are suspect. How can you maximize the sum of the correlations of the components without any orthogonality/uncorrelatedness constraint? Is there anything then stopping the components from all being equal, specifically all representing the first canonical correlation? That is what the uncorrelatedness constraint is there to prevent.
3) Eq. (5) is not clear to me. How does this represent the Wasserstein distance? Can you explain how this adversarial learning achieves what you claim?
4) In many places, "constraints" are discussed (e.g., "we constrain the latent representations for each of the two modalities to follow the same distribution") but as far as I can tell these are not really constraints; they take the form of penalties in the joint objective.

**Summary Of The Paper:**

A method is proposed for learning correlated representations of two data views that purports to be superior to existing approaches in terms of correlation captured, utility of representations for classification, and training efficiency (since stochastic optimization can be used).

**Summary Of The Review:**

There may be some value to the proposed method, but the motivation is suspect and the paper is less than clear in several places.

---

> ### Author Response · Authors · 2022-11-14
> **Improve clarity, derivation details, and provide time structure analyses**
>
>
> We greatly appreciate the reviewer’s constructive suggestions, which have dramatically helped improve the clarity of our work. We addressed the reviewer’s concerns below and revised the manuscript accordingly.
> ## Reply to point 1
>
> In a general multimodal framework, the total loss includes loss terms from all of the multimodalities, such as reconstruction errors from modality x and y. If two modal features have dramatically different distributions or feature values, without proper balancing of their contributions, optimization can bias towards one of the modalities.
>
> In existing approaches, the standard preprocessing approach is to perform mean-centered transformations before feeding the data into the CCA model. Then, within the CCA models, modalities are treated equally. However, this can perform poorly if the two modal features have different distributions. We illustrate in the manuscript how existing methods suffer from unbalanced modalities by (Section 3.1.2, Figure 4): starting with balanced distributions, transforming the features in one modality using multi-log operations or binary hashing, re-running the CCA approaches, and showing that the performance degrades.
>
> In our proposed model adCCA, we avoid the problem of unbalanced modalities by: 1) modality-specific autoencoders that are optimized independently; and 2) latent representations that are aligned indirectly using an adversarial discriminator.
>
> Thank you for pointing out that this was unclear and we have rerevised the accordingly.
>
> ## Reply to point 2
>
> Thank you for pointing out that this was unclear. In the standard CCA, orthogonality is achieved through SVD decomposition. That is why CCA requires optimization through total samples. In our model, to enable optimization using the standard mini-batch paradigm, we instead use representations from the autoencoder bottleneck layer and a regularization penality to enforce correlation. The non-equal and uncorrelated properties of the two modality representations are provided by the two independent autoencoders.
>
> We appreciate the reviewer for pointing this out. We have revised the explanation in Corollary 1 and explicitly added the penalty term. Please refer to Page 3. (Theorem 1 is not applicable because it is on single canonical variable).
>
> ## Reply to point 3
>
> We apologize for the confusion. In the revised manuscript, we added the full formulation of the Wasserstein distance between latent distributions as well as derivations to derive an adversarial target that minimizes Wasserstein distance. Please refer to new Eqs. 5-7.
>
> ## Reply to point 4
>
> Thank you. We now use your suggested term of “penalties”.
>
> ## Reply to [Difference between CCA variants]
>
> To better highlight novelty, we now provide a table summarizing the differences in representation learning and training between our approach and existing CCA methods (Appendix Table F.6). Standard CCA (e.g., CCA and kernel CCA) learns a canonical representation for each modality but requires the optimization over full dataset. As a comparison, most recently developed deep learning CCA approaches try to combine multimodalities and learn a single joint representation. This joint representation learning is different from the original CCA learning target, though it enables training on mini-batches. Our approach follows the classical CCA target and is optimizable with standard mini-batch training.
>
> ## Reply to [Training time]
>
> We provided the training time result for each tested method in Appendix Figure D.4. The runtime shows adCCA is shower than VCCA, DPCCA and simCLR due to its multi-step training, but it is faster than Kernel CCA and Deep CCA.
>
> ## Reply to [Model structure analysis]
>
> We appreciate the reviewer for the suggestion to perform model structure analysis. In the revision, we considered the consequences of removing the AE initialization, Wasserstein distance term, or inner product term and re-ran the adCCA model. From the evaluation (Appendix Table F.1),
>
> 1. Removing the AE initialization did not affect the correlation result because of the strong penalty from Wasserstein distance and inner product; however, it reduced the label prediction accuracy because of the imperfect bottleneck embedding.
> 2. Removing the Wasserstein term caused the representations from the two modalities to have different latent distributions. As expected, this greatly reduced correlation.
> 3. Removing the inner product term also compromised the correlation result. The Wasserstein distance loss alone only encourages the overall latent distributions from the two modalities to be the same, but does not require the two latent representations for the same sample to be matched.
>
> Taken together, all tested components are important for canonical representation learning.
>
> We also take the reviewer’s concern about clarity to heart. In addition to all the above changes, we added a new Appendix Figure D1, which directly highlights the motivation and novelty of this work.

---

### Decision · Program_Chairs · 2023-01-20

**Decision:**

Reject

**Justification For Why Not Higher Score:**

The reviewers found common issues for the paper, and I have additional concerns (discussed in the meta review).

**Justification For Why Not Lower Score:**

N/A

**Metareview: Summary, Strengths And Weaknesses:**

This work derived a deep adversary CCA method, with the intuition/assumption that representations of different modalities obey an identical distribution in a common latent subspace.

Strengths:
The authors had a reasonably good survey of literature and some experimental study.

Weakness:

Most reviewers complained about the writing being not precise and some claims not supported. As mentioned by reviewers, having a mini-batch training procedure does not make a significant contribution as it was achieved by DCCA already. In my opinion, the key difference by this paper is to simultaneously minimize distribution-wise difference in latent space representations, while maximizing sample-wise correlation. It is not very clear whether theses two terms are contradictory or complimentary to each other, and this point should probably be investigated thoroughly. I also have concerns on the two-step approach where a first auto-encoding step is used to extract features for correlation based learning: unless the auto-encoding step is loss-less,  correlated dimensions that are not contributing significantly to reconstruction are already lost in that step.